# High-Speed Videoendoscopy Enhances the Objective Assessment of Glottic Organic Lesions: A Case-Control Study with Multivariable Data-Mining Model Development

**DOI:** 10.3390/cancers15143716

**Published:** 2023-07-22

**Authors:** Jakub Malinowski, Wioletta Pietruszewska, Konrad Stawiski, Magdalena Kowalczyk, Magda Barańska, Aleksander Rycerz, Ewa Niebudek-Bogusz

**Affiliations:** 1Department of Otolaryngology, Head and Neck Oncology, Medical University of Lodz, 90-419 Lodz, Poland; jakub.tomasz.malinowski@umed.lodz.pl (J.M.); magdalena.kowalczyk@stud.umed.lodz.pl (M.K.); magda.baranska@stud.umed.lodz.pl (M.B.); ewa.niebudek-bogusz@umed.lodz.pl (E.N.-B.); 2Department of Radiation Oncology, Dana-Farber Cancer Institute, Harvard Medical School, Boston, MA 02115, USA; konrad.stawiski@umed.lodz.pl; 3Department of Biostatistics and Translational Medicine, Medical University of Lodz, 90-419 Lodz, Poland; aleksander.rycerz@stud.umed.lodz.pl

**Keywords:** glottis organic pathology, glottic cancer, high-speed videoendoscopy, kymography, machine learning

## Abstract

**Simple Summary:**

The standard protocol for distinguishing between benign and malignant lesions remains clinical judgment and histopathological confirmation by an experienced otolaryngologist. An additional tool is high-speed videoendoscopy (HSV), an accurate method for an objective assessment of vocal fold oscillations. The aim of the study was to utilize a quantitative assessment of the vibratory characteristics of vocal folds in diagnosing benign and malignant lesions of the glottis using HSV. The machine learning model identifying malignancy among organic lesions reached an AUC equal to 0.85 and presented with 80.6% accuracy, 100% sensitivity, and 71.1% specificity on the training set. Important predictive factors were frequency perturbation measures. The results suggested that advanced machine learning models based on HSV analysis could potentially indicate a heightened risk of cancerous mass. Therefore, this technology could, in future, aid in early cancer detection; however, further investigation and validation is needed.

**Abstract:**

The aim of the study was to utilize a quantitative assessment of the vibratory characteristics of vocal folds in diagnosing benign and malignant lesions of the glottis using high-speed videolaryngoscopy (HSV). Methods: Case-control study including 100 patients with unilateral vocal fold lesions in comparison to 38 normophonic subjects. Quantitative assessment with the determination of vocal fold oscillation parameters was performed based on HSV kymography. Machine-learning predictive models were developed and validated. Results: All calculated parameters differed significantly between healthy subjects and patients with organic lesions. The first predictive model distinguishing any organic lesion patients from healthy subjects reached an area under the curve (AUC) equal to 0.983 and presented with 89.3% accuracy, 97.0% sensitivity, and 71.4% specificity on the testing set. The second model identifying malignancy among organic lesions reached an AUC equal to 0.85 and presented with 80.6% accuracy, 100% sensitivity, and 71.1% specificity on the training set. Important predictive factors for the models were frequency perturbation measures. Conclusions: The standard protocol for distinguishing between benign and malignant lesions continues to be clinical evaluation by an experienced ENT specialist and confirmed by histopathological examination. Our findings did suggest that advanced machine learning models, which consider the complex interactions present in HSV data, could potentially indicate a heightened risk of malignancy. Therefore, this technology could prove pivotal in aiding in early cancer detection, thereby emphasizing the need for further investigation and validation.

## 1. Introduction

The increasing demands for communication make a healthy voice a crucial determinant for good quality of life in contemporary society [1,2]. Voice production is a complex process, mainly including the glottis, where vocal folds vibrate synchronically to produce the sound [3]. Therefore, an assessment of the phonatory function is one of the key targets for the appropriate diagnosis and treatment of glottic pathologies. Concurrently, dysphonia is the most common problem encountered by laryngologists. Patients usually complain about altered vocal quality, hoarseness, abnormal pitch, change in timber, weak or tremulous voice, and vocal fatigue. Other signs are a long-term worsening of voice efficiency and total lack of voice (aphonia). The causes of dysphonia of different severity levels are functional and/or organic pathologies including benign and malignant lesions. The first type of lesion forms a heterogeneous group containing, among others, Reinke’s edema, vocal nodules, polyps, and granulomas. They often differ in their elasticity, mass, and size, with various impacts on glottal vibrations. On the other hand, malignant lesions, even at an early stage, increase vocal fold stiffness and mass, limit or abolish the mucosal wave, and may reduce vocal fold mobility.

Over the years, methods for imaging vocal function have developed extensively [4]. As of today, the most widespread technique, often held as the ‘gold standard’, remains laryngovideostroboscopy (LVS) [5,6,7]. However, it has certain limitations originating in the technical aspects of acquiring images. LVS often fails in patients with irregular or short vocal cycles due to problems with the full synchronization of the strobe light with the frequency of the vocal fold vibrations. The resultant image of irregular vocal fold vibrations makes the assessment of certain vibratory features impossible, as in static light endoscopy [8,9]. These problems often occur in patients with advanced glottic lesions, resulting in there being no tool to assess vocal fold vibratory patterns in this group. Moreover, LVS is highly dependent on the examiner’s level of experience. Thus, new technologies involving the use of high-speed video cameras are increasing in popularity and interest among clinicians [10,11].

High-speed videoendoscopy (HSV), due to the constant developments in recent years, can now be applied not only in research setup but also in everyday clinical practice [12,13,14,15]. Until recently, it was mainly used in the assessment of normophonic voices and functional phonation disorders [12,16,17]. However, it allows for a slow-motion video of vocal fold vibrations in both short and asynchronous phonations to be obtained. This is a strong technical advantage in severe voice disorders caused by glottic lesions [18,19]. High-quality recordings containing numerous cycles form a solid basis for generating kymographic sections and applying various digital analysis algorithms, allowing for an objective evaluation of phonatory function, including an assessment of parameters describing the regularity and amplitude of vibrations [5,20,21,22].

Such a detailed, objective analysis enables the detection of vocal function disorders that are unnoticeable during an examination with other techniques, because some discreet disruptions can be revealed only after analysis of thousands of vocal cycles [23,24]. As it is crucial to differentiate between benign and malignant lesions, efforts are being made to obtain this goal with non-invasive methods.

The aim of the study was to quantitatively assess the vibratory characteristics in benign and malignant lesions of the vocal folds using high-speed videoendoscopy. We use advanced artificial intelligence tools to develop and validate models that can assist clinicians in the detection of any organic glottic changes and differentiation between malignant and benign lesions.

## 2. Materials and Methods

### 2.1. Study Group

The research included 138 patients. The study group consisted of 100 patients hospitalized at the Department of Otolaryngology, Head and Neck Oncology, of the Medical University of Lodz, with dysphonia and unilateral lesions on the vocal fold. The control group was composed of 38 normophonic subjects without vocal fold pathology on clinical examination who had been hospitalized for other otolaryngological conditions. Inclusion criteria for the study group were: ≥18 years of age, dysphonia, and the presence of a unilateral organic glottic lesion. The inclusion criteria for the control group were: ≥18 years of age, no history or current dysphonia, and no structural or functional abnormalities in the larynx. The exclusion criteria for both groups were the inability to obtain recordings that were suitable for analysis, and organic lesions involving both sides of the larynx.

Among subjects with dysphonia, 64 were diagnosed with benign vocal lesions (specifically: 50 polyps and 14 vocal cysts). This group consisted of 41 women and 23 men, aged from 22 to 74 years; the average patient age was 53.62 ± 12.79 years. The remaining 36 subjects were diagnosed with early glottic cancer—out of them, 27 were male and 9 were female, aged from 42 to 76 years, with an average age of 67.22 ± 6.66 years. All of the organic lesions were confirmed by histopathologic examination. All malignancies were Grade 1 (well differentiated) or Grade 2 (moderately differentiated) invasive squamous cell carcinoma, in some cases accompanied by high-grade dysplasia. Clinically, the lesions were either exophytic tumors or infiltrative growths encompassing one vocal fold, classified as cT1. The control group comprised 38 normophonic subjects without vocal fold pathology on clinical examination, who had been hospitalized for other otolaryngological conditions. In the control group, there were 25 women and 13 men, aged from 19 to 83 years, with an average age of 45 ± 18.23 years. For the purpose of further analysis, all subjects were divided into three groups: normophonic, subjects with benign lesion and subjects with malignant lesion.

Approval for this study was granted by the Ethical Committee of the Medical University of Lodz (no. RNN/96/20/KE 8 April 2020), and all patients provided their written informed consent to participate in the research.

### 2.2. Equipment and Examination

First, all patients were assessed by an ENT specialist. The examination included a complex interview concerning laryngeal symptoms. Each subject was required to complete voice quality questionnaires, as monitoring patients’ functional outcomes is of the utmost importance in laryngological practice; it can also facilitate the quicker detection of voice disorders [25]. For this purpose, we used validated surveys, including the Voice Handicap Index (VHI), Voice-Related Quality Of Life (V-RQOL), Vocal Tract Discomfort Scale (VTDS), and Voice Fatigue Index (VFI). Next, all patients underwent a larynx examination with the rigid, oval endoscope Fiegert-Endotech ϕ12.4/7.2 equipped with a high-speed camera (ALIS Cam HS-1; DiagNova Technologies, Wrocław, Poland; version 1.3), which can record 2400–3200 frames per second (fps). Two recording modes were applied. Firstly, an initial visualization and assessment of the larynx were conducted, with the camera operating at around 24 fps. This allows for the initial identification of pathologies and is used to center the camera sensor on the glottis and adjust the focal length of the camera optic systems (image sharpness is adjusted semi-automatically). Subsequently, the high-speed mode was used, capturing images from the central part of the sensor. For the purpose of the study, images were recorded at a rate of 3200 fps. The length of the high-speed recording was 2000 frames, resulting in a recording time of 625 ms. More detailed technical aspects of high-speed recording can be found in our previous publication [25].

### 2.3. Kymographic Analysis

The recorded images were later subjected to a detailed kymographic analysis, using dedicated software. Based on the kymographic section, first, the edges of the vocal folds were identified automatically. Next, the glottal width waveform (GWW) was generated—a graph of glottal width changes in the function of time. In the case of organic glottic lesions, GWW was used, which is able to better visualize local changes in the function of the vocal folds, which are directly related to the presence of pathological mass [26,27]. This represents vocal fold movements reflecting instantaneous changes in the glottal area at different glottal length levels for the visualization of glottal dynamics.

For the GWW, a kymographic cross-section in the glottis’s middle part was generated, which proved the location of the largest vibration amplitude and had the biggest impact on voice generation and its pathologies [28]. It was previously recommended as the preferred location for kymographic analysis [29].

Next, based on the GWW graph, each vocal fold’s oscillation was identified and quantitatively analyzed. In this evaluation, information about temporal aspects of vocal fold dynamics can be gathered, representing the impact of organic lesions on the periodicity of glottal vibrations. Aside from the fundamental frequency of vocal fold vibration, the focus wa on temporal parameters, because they tend to be more sensitive in the evaluation of organic lesions, as observed previously [30]. Two main groups of parameters were determined: the Jitter group, which describes changes in the frequency of oscillations and includes period perturbation measures: mean Jitter value given in % (Jitt), mean Jitter value (Jita), Period Perturbation Factor (PPF), Period Relative Average Perturbation (PRAP), Period Perturbation Quotient with an average of 3 vocal cycles (PPQ3) and Period Perturbation Quotient with an average of 5 vocal cycles (PPQ5), and the Shimmer group, parameters related to changes in the amplitude of vocal cycle, called amplitude perturbation measures: Shimmer, Amplitude Perturbation Factor (APF), Amplitude Relative Average Perturbation (ARAP), APQ3 (Amplitude Perturbation Quotient with an average area of 3 vocal cycles (APQ3) and Amplitude Perturbation Quotient with an average area of 5 vocal cycles (APQ5). The parameters were chosen because of their proven clinical usefulness. Details about the analysis and calculation of individual parameters were described in our previous publication [26].

### 2.4. Statistical Analysis

Statistical analyses were performed using RStudio 4.2.0. First descriptive statistics were calculated. The threshold for statistical significance was set to *p* < 0.05. Differences between groups were tested using the non-parametric ANOVA–Kruskal–Wallis test. Subsequently, receiver operating characteristic (ROC) curves for individual parameters were plotted.

For modeling purposes, the dataset was randomly divided into training (70%) and testing sets (30%) (Table 1). To assess whether HSV-derived features can differentiate patients with organic lesions and healthy patients, as well as differentiate between malignant and benign lesions, we performed feature selection and modeling using OmicSelector software for both problems [31]. In brief, a standard set of feature selection methods returned sets of candidate features, which were further benchmarked using selected predictive modeling methods. The final set of features was chosen based on mean testing accuracy. In the next step, the final models were developed using extreme gradient boosting (XGBoost) with Bayesian optimization of hyperparameters. The accuracy, sensitivity, and specificity of developed models were assessed in training and testing sets. OmicSelector environment utilized R (version 4.2.1).

In summary, using the same data split and approach, we developed separate models for the detection of organic lesions and differentiation of malignant and benign lesions. Additionally, we assessed the influence of variables on the final models’ predictions using Shapley (SHAP) values [32]. The models analyzed all the individual parameters and their interdependencies. The multivariable approach ensured that there was need for dataset stratification according to gender.

## 3. Results

### 3.1. HSV Images

The patients from each group (normophonic, subjects with benign lesion and with malignant lesion) were selected for the analysis method (Figure 1).

### 3.2. Objective Analysis—Parameters

The quantitative parameters describing vocal fold oscillations were compared between the following groups: normophonic (healthy), subjects with benign and with malignant lesions (Table 2). All 12 parameters, aside from average fundamental frequency (F0Avg), differed between the study and control cohort in a statistically significant way. Both the Jitter and Shimmer group parameters were significantly increased in patients with glottic organic lesions compared to the control group (Table 2; Figure 2). However, F0Avg was non-significantly lower in the malignant lesion subjects. There were no such relations between the benign and malignant lesion groups (Figure 2).

The plotted ROC curves for individual parameters indicated the highest values of the area under the curve (AUC) for the differentiation of normophonic subjects vs. patients with any organic lesions. Parameters such as Jita (AUC = 0.739; 95%CI: 0.646–0.832), APQ3 (AUC = 0.733; 95%CI: 0.644–0.821) and ARAP (AUC = 0.731; 95%CI: 0.642–0.819) were the best diagnostic factors for organic lesions when compared with normophonic subjects (Figure 3A–C). When differentiating between malignant and benign lesions, the strongest discriminating factors were parameters of frequency perturbation (Jitter group): PPF (AUC = 0.613; 95%CI: 0.499–0.728), Jitt (AUC = 0.612; 95%CI: 0.497–0.726) and Jita (AUC = 0.596; 95%CI: 0.479–0.712) (Figure 3D–F). Shimmer group parameters achieved lower values. However, for the whole group, the AUC values were low (*p* = 0.0006). The results of our ROC analysis are included in the Appendix A.

After determining that no single parameter was able to distinguish normophonic subjects and those with benign and malignant lesions with acceptable accuracy, a multivariable predictive model was developed. After feature selection, the final model for diagnosis of the presence of an organic lesion (vs. normophonic subjects) utilized F0Avg, Jitt, PPQ3, PPQ5, APQ3, and gender as predictors (Figure 4). As mentioned before, the model analyses values of individual parameters, on their own and in relation to other parameters, were used to create a decision tree, finally leading to a result determining the probability of the presence of any organic lesion. Due to the inclusion of gender and fundamental frequency among the parameters, analysis of both men and women with the same algorithm was possible (without splitting them into separate groups). The best model achieved AUC equal to 0.983 (95%CI: 0.9665–1.0000) and presented perfect accuracy on the training set and 89.3% accuracy on the testing set. The sensitivity of this model on the testing set was 97.0% and the specificity was 71.4%.

Although extreme gradient boosting models tend to analyze complex interactions between predictors, the most significant feature of this model was F0Avg and APQ3. Increasing values of F0Avg reduced the probability of organic lesions of the vocal folds. The high value of APQ3 increases the chance that the patient has any organic glottic pathologies. APQ3 is one of the parameters from the Shimmer group that determines the amplitude perturbation of phonatory oscillations. The least significant features for the differentiation between organic lesions of glottis and healthy glottis are gender and PPQ5.

Next, a model aiming to identify malignancy among organic lesions was developed (Figure 5). For differentiation between cancerous and benign lesions, the best set of predictors was F0Avg, Jitt, Jita, APF, Shimmer, PRAP, PPQ3, PPQ5, APQ5, and gender. The model utilizing those predictors achieved an AUC equal to 0.85 (95%CI: 0.78–0.92; (Figure 5)) and presented with 80.6% accuracy, 100% sensitivity, and 71.1% specificity on the training set. On the testing set, this model achieved 78.8% accuracy, 100% sensitivity, and 63.2% specificity. Again, in this model, F0Avg showed the most prominent influence on predicted validation; however, a higher F0Avg increased the chance of malignancy. The second most important factor for the algorithm was gender—males indicated an increased risk of cancerous lesions of the vocal folds. The third most important factor was Jita, a parameter describing frequency perturbation, and its higher values increased the risk of malignancy. Parameter PPQ3 seems to have a marginal effect on the model’s prediction, however, the elimination of this feature leads to inferior performance. Both models are included in Appendix A. RDS files can be loaded in R software (version 4.2.1) and OmicSelector package can be used for model scoring on new cases. Most important in this case was maintaining high sensitivity so that no case of cancer was omitted.

## 4. Discussion

In this research, the authors performed the first and the largest study using HSV to pre-operatively differentiate organic lesions based on the temporal parameters describing vocal fold oscillations [27,33,34]. The accurate diagnosis and management of glottic lesions depends on a precise preoperative visualization of the glottal structure and an objective assessment of phonation. High-speed videoendoscopy was mostly used in functional disorders but has recently gained importance in the assessment of organic glottic lesions [19,23,35].

HSV, as opposed to LVS, can capture even short and asynchronous vocal fold oscillations, which are common in patients with severe organic lesions [23]. Due to its high sampling speed, HSV shows actual oscillations in real-time in consecutive cycles, which facilitates the detection of both discreet and intermittent disturbances in vocal fold vibrations. Hence, several clinicians prefer to use HSV over LVS in the objective assessment of organic lesions, using a variety of methods [22,36,37]. Furthermore, various disorders display disease-specific vibratory disturbances that can help in the differential diagnosis [21,36]. Powell et al. compared organic-based pathological voice vibrations in LVS and HSV—more patients were accurately evaluated in the latter. Additionally, more parameters markedly changed in HSV than in LVS in the post-operative examination [18]. The contribution of the authors to the development of this scientific trend is significant. We add to the literature by combining a qualitative assessment with parametrization and quantitative analysis to evaluate the structure of the lesion and its influence on phonation. Disturbances in glottal oscillations caused by organic lesions are varied and depend on the morphology and infiltration of the vocal fold of the pathological mass [35]. Benign masses cause smaller irregularities and changes in the amplitude of oscillations, mucosal wave, and phase asymmetry because they are predominantly soft and elastic, and do not infiltrate the vocal fold. We present this in Subject 2, in which vocal fold with benign polyp preserved both vibratory capability and mucosal wave—marginally modeled by the lesion, causing a minimal disturbance in oscillation frequency and increasing amplitude [35]. Stiff, heavy, and infiltrative malignant masses cause larger disruptions in the vocal cycle. This is visible in Subject 3, in which cancer lesions involved the whole vocal fold—vibratory amplitude and mucosal wave were absent. In the parametrical analysis of the oscillations, we observed a heavily disturbed frequency and amplitude. Similarly, Gandhi et al. performed a qualitative assessment of glottic carcinoma, both pre-and postoperatively [38]. Other researchers also proved that HSV has potential for clinical adaptation; however, it lacks standardized investigation protocol and analysis tools [34,39,40].

We aimed to create, and were successful in creating, such a protocol by combining a qualitative assessment of HSV recordings with parametric analysis acquired from HSV kymography. We applied the kymographic section to generate a glottal width waveform (GWW). Based on the GWW plot, we calculated the fundamental frequency (F0) and temporal parameters, describing changes in the oscillation of the vocal folds—the regularity, periodicity, and amplitude of vibrations. In particular, differences in temporal parameters between healthy subjects and patients with organic pathologies are more substantial than those in spatial parameters [19,30]. In our study, all temporal parameters differed significantly between healthy subjects and patients with organic pathologies, similar to the literature [40]. As for Jita—a parameter describing changes in the frequency of oscillations—it differed statistically significantly between healthy subjects and patients with organic pathologies and reached the highest AUC as an individual parameter. Similarly, Shimmer—expressing the amplitude of vocal fold oscillations and, therefore, phonation stability—also differed significantly in groups with a high AUC.

Furthermore, we analyzed if the HSV-derived parameters can differentiate between malignant and benign lesions, but none of the parameters appeared to be a significant predictor. However, using a complex data-mining approach, we developed extreme gradient boosting models that achieved clinically acceptable performance metrics in hold-out validation. Our paper follows the state-of-the-art approach to predictive model development [41]. First, we conducted an extensive feature selection on the training set to limit redundancy and potential multicollinearity. Using hold-out validation, we tested generalizability and limited the chance of overfitting. To model induction, we used extreme gradient boosting, which was proven to be superior for tabular data [42]. Lastly, through the application of SHAP values, we tried to explain the models’ behavior, which is in concordance with the concept of explainable machine learning [43]. We included our models as Appendix A to facilitate potential clinical implementation.

In the literature, men and women are assessed separately, as there are large observed differences in quantitative vocal fold vibration parameters between males and females [19,30]. The predictive model used in our research included gender in both best feature sets, and thus was able to take into account its impact on fundamental frequency and other temporal parameters. As a result, we were able to analyze both men and women with a single model, making splitting groups by gender unnecessary.

In our model, the most significant features for the detection of the normophonic subjects and patients with any organic lesions were F0Avg and APQ3. Increasing values of F0Avg reduced the probability of hypertrophic pathologies of the glottis. The mass of organic lesion weighing down the vocal folds may cause them to vibrate with a lower frequency. High APQ3 values increase the chance that the patient has any organic lesions. An increase in the involved vocal fold mass may impair the ability to reach the maximum amplitude of oscillations, causing fluctuations in the stability of phonation.

There are few reports in the literature studying whether HSV can assess the difference between malignant and benign vocal fold lesions; for this reason, we developed another model [22,36]. Based on the laryngotopographic analysis of HSV, Yamauchi et al. indicated that the presence of a non-vibrating area was observed in glottic cancer [36]. Thus, in our previous study concerning a laryngotopographic evaluation of HSV, we proposed the Stiffness Asymmetry Index, comparing quantitatively healthy and organically affected vocal folds [44]. However, in this study, we included unilateral organic glottic lesions and used temporal parameters. The best predictors for the model were F0Avg, Sex, Jita, APQ, resulting in an AUC of 0.85 (95%CI: 0.78–0.92). This model showed that higher values of F0Avg increase the risk of malignancy. Unlike in the previous model, this observation cannot be explained by the effect of mass. We consider that it may be caused by more tense and forced phonation, with a higher fundamental frequency in subjects with malignant glottic lesions in comparison with patients with benign lesions. These patients struggled to achieve sustained phonation at a comfortable level of pitch and loudness during the examination with rigid optics. Another finding of the study is that higher values of Jita increased the risk of malignancy. This implies that a stiff malignant mass severely disrupts the regularity of glottal oscillation, in comparison to soft, benign lesions.

Our research has some limitations. Although we tried to provide a possible explanation for the meaning of individual parameters for our model, we are aware that predictive model behavior is difficult to explain because of its complex data-mining algorithms. Clinical interpretation of the second model is especially intricate because of the mutual interactions between the parameters, which is a general problem in studies that use machine learning [43,45].

To summarize, we proved the value of differentiate benign and malignant lesions with temporal vocal fold oscillation parameters, calculated on the basis of a kymographic analysis of HSV recordings. We believe that an objective HSV analysis of vocal function with the employment of machine learning may play an important clinical role. It can assist less experienced physicians who may have difficulties in differentiating certain benign and malignant lesions. A quantitative approach in this case is crucial, as subjective kymographic interpretation causes many difficulties, resulting in low interrater agreement. In the foreseeable future, the most important part of the diagnostic and treatment process will be examination by experienced ENT specialists, although we believe that HSV can also be of assistance, especially in doubtful cases, by supplying the aforementioned objective parameters. It may also be useful while planning the single-stage transoral laser microsurgery approach to remove the suspected cancerous lesion in a single procedure, omitting the need for pre-operative biopsy and possible complications including tissue inflammation and fibrosis [46]. The presented machine learning approach is a promising method to create a useful supporting tool; however, further research is needed.

Research on a larger population would minimize any bias resulting from the insufficient number of participants, broaden the scope of our studies and quantitatively assess bilateral lesions and their dynamic behavior. We believe that, with the further development of algorithms and improvements in the use of SHAP values, it may be possible to explain the data-mining models.

## 5. Conclusions

In conclusion, our study provides a comprehensive analysis of the predictive capabilities of audio and HSV data in the assessment of vocal fold oscillations. The standard protocol for distinguishing between benign and malignant lesions continues to be clinical evaluation with videolaryngoscopy, executed by an experienced ENT specialist and finally confirmed by histopathological examination. Our study did not establish that any single parameter derived from HSV kymographic analysis could differentiate noncancerous from malignant laryngeal lesions. However, our findings did suggest that advanced machine learning models, which consider the complex interactions present in HSV data, could potentially indicate a heightened risk of malignancy. Therefore, this technology could prove pivotal in aiding in early cancer detection, thereby emphasizing the need for further investigations and validations.

## Figures and Tables

**Figure 1 cancers-15-03716-f001:**
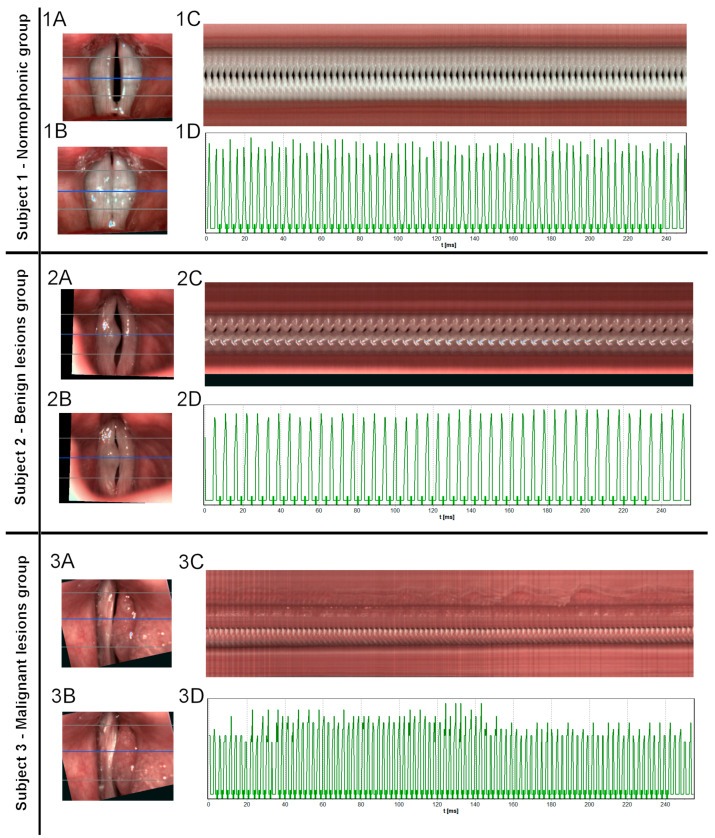
Images of the glottis for Subjects 1, 2, and 3 obtained by HSV recording. Letters indicate each stage of the analysis. (**A**)—images of vocal folds in the opening phase (**B**)—images of vocal folds in the closing phase. In images A and B, horizontal lines mark three cross-sections, while a blue line marks the middle cross-section used later for generating videokymogram. (**C**)—videokymogram obtained from the middle part of the glottis. (**D**)—glottal width waveform (GWW) plot obtained from videokymogram shown above. Subject 1: 25-year-old female, nonsmoker, with no history of dysphonia. During the laryngological examination, no functional or organic pathology was found in the larynx (**1A**). HSV did not reveal any significant irregularities in the frequency of vocal vibrations and did not change the diagnosis. (**1C**) However, a slight tendency to hyperphonation was observed, accompanied by a small accumulation of mucus found in the middle part of the glottis. In the closing phase, the glottis reached almost full closure, with slight insufficiency in the posterior part, which is very often observed in females (**1B**). The GWW graph revealed slight irregularity in the amplitude of vocal vibrations, which could be explained by the tendency to hyperphonation. (**1D**) Subject 2- a 55-year-old male who complained of mild dysphonia. Laryngological examination revealed a hypertrophic lesion in the form of a sessile polyp on the medium one-third of the right vocal fold (**2A**,**B**). The post-surgery histopathologic examination confirmed the initial diagnosis of the benign polyp. HSV imaging revealed that the involved vocal fold has vibratory capability. The mucosal wave was preserved on both vocal folds, however, it was slightly modeled by the lesion. (**2C**) The increased mass of this right vocal fold resulted in aperiodic oscillation and occasional phase difference between the two vocal folds. GWW graph revealed minimal disturbance in the frequency of oscillations and slightly higher in the amplitude (**2D**)—leading to mild severity of dysphonia. Post-operative histopathologic results confirmed the initial diagnosis. Subject 3–69-year-old male patient, a smoker, with a history of severe dysphonia. During the laryngological examination, a large irregular hypertrophic mass of the total length of the left vocal fold was found and initially assessed as a lesion with a high risk of malignancy. (**3A**,**B**) Based on the HSV recording the mass was assessed as stiff. Neither vibratory amplitude nor mucosal wave was present on the involved fold indicating infiltration of deep tissues (**3C**). The high mass of the lesion caused disruption in oscillations of the healthy contralateral vocal fold. In this case, a biopsy was performed revealing planoepithelial carcinoma Grade 2. The GWW graph shows a high grade of irregularity both in frequency and amplitude of oscillations (**3D**).

**Figure 2 cancers-15-03716-f002:**
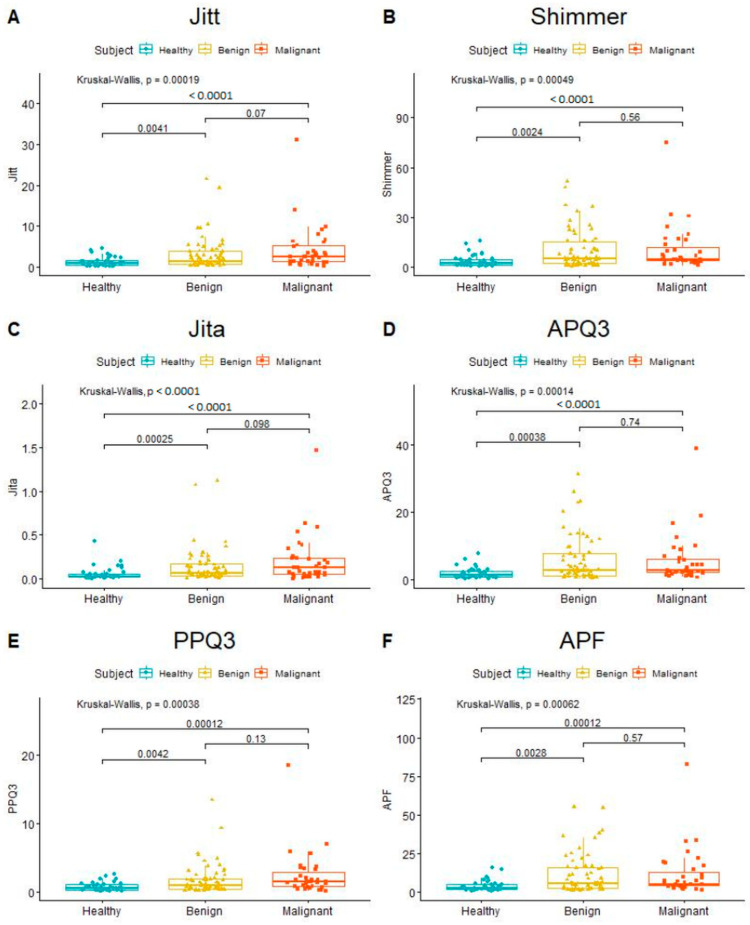
Diagrams showing the results of the Kruskal–Wallis test comparing values of chosen individual parameters between healthy, benign, and malignant subjects. In the left column (**A**,**C**,**E**)—Jitter group parameters describing frequency irregularities. In the right column (**B**,**D**,**F**)—Shimmer group parameters describing amplitude irregularities.

**Figure 3 cancers-15-03716-f003:**
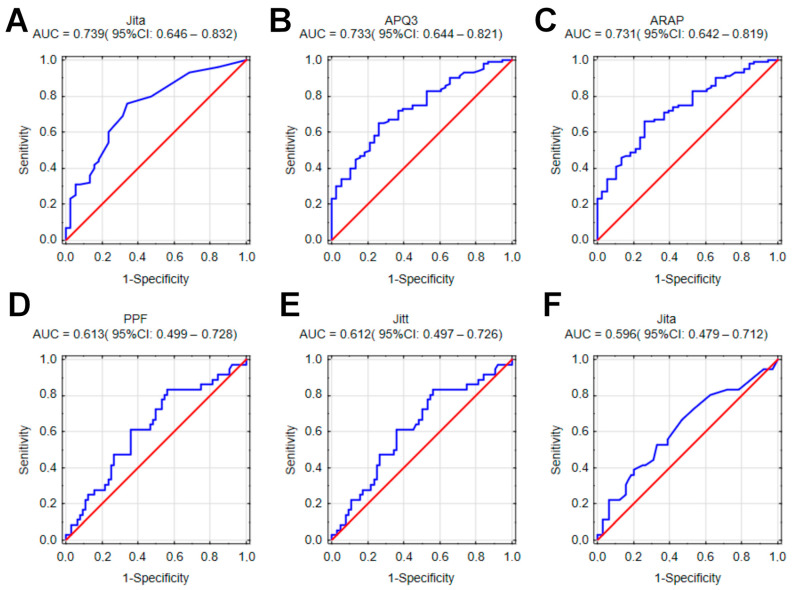
ROC curves plotted for individual parameters in differentiating of subjects in groups—as described in the text. (**A**) Jita—normophonic subjects vs. any organic lesion; (**B**) APQ3—normophonic subjects vs. any organic lesion; (**C**) ARAP normophonic subjects vs. any organic lesion; (**D**) PPF– benign vs. malignant; (**E**) Jitt—benign vs. malignant; (**F**) Jita—benign vs. malignant.

**Figure 4 cancers-15-03716-f004:**
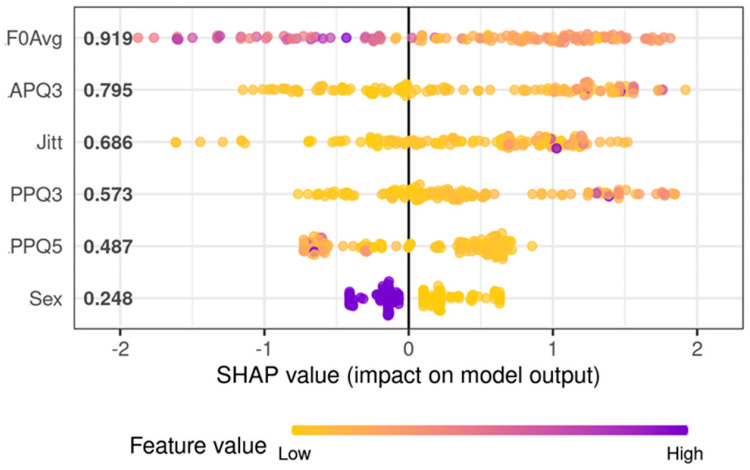
SHAP values showing the impact of individual vocal fold vibration parameters on model output result. Higher SHAP values indicate the larger influence of parameters on the final result. The panel shows the predictors for the final model used for the diagnosis of organic glottic lesions.

**Figure 5 cancers-15-03716-f005:**
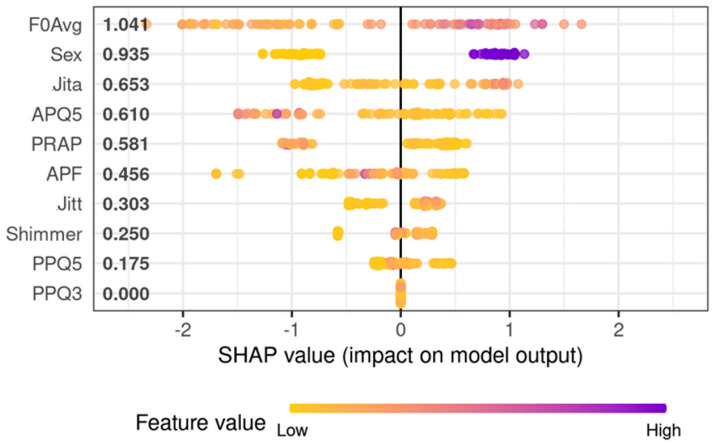
SHAP values showing the impact of individual vocal fold vibration parameters on model output result. Higher SHAP values indicate an increased influence of parameters on the final result. The panel presents SHAP values for predictors used in the model to discriminate malignant and benign tumors of the glottis.

**Table 1 cancers-15-03716-t001:** The number of samples selected from all examined subjects and randomly assigned to training (70%) and testing (30%) sets for the multi-factor analysis models. The split presented no bias associated with the class (Chi2 test, *p* = 0.88).

Group	Training Set (n = 99)	Testing Set (n = 39)
Normophonic subjects	27	11
Subjects with benign lesion	45	19
Subjects with malignant lesion	27	9

**Table 2 cancers-15-03716-t002:** The comparison of quantitative parameters describing vocal fold vibrations in normophonic subjects and patients with benign and malignant glottic lesions. Kruskal–Wallis test, median values.

Parameter	Normophonic Subjects	Subjects with Benign Lesion	Subjects with Malignant Lesion	*p*-Value
F0Avg [Hz]	259.04 ± 107.33	223.2 ± 78.04	224.77 ± 72.99	*p* = 0.1700
Jitt [%]	1.3 ± 1.14	3 ± 3.99	4.26 ± 5.51	*p* = 0.0002
Jita [ms]	0.06 ± 0.08	0.14 ± 0.2	0.21 ± 0.27	*p* < 0.0001
PPF [%]	1.29 ± 1.1	2.96 ± 3.93	4.24 ± 5.42	*p* = 0.0002
PRAP [%]	0.72 ± 0.64	1.71 ± 2.28	2.3 ± 3.18	*p* = 0.0004
PPQ3 [%]	0.71 ± 0.62	1.69 ± 2.25	2.32 ± 3.21	*p* = 0.0004
PPQ5 [%]	0.75 ± 0.73	1.74 ± 2.6	2.32 ±3.6	*p* = 0.0008
Shimmer [%]	3.85 ± 3.55	10.21 ± 11.87	10.81 ± 5.14	*p* = 0.0005
APF [%]	3.89 ± 3.62	10.78 ± 12.91	11.39 ± 14.87	*p* = 0.0006
ARAP [%]	1.81 ± 1.63	5.5 ± 6.42	5.36 ± 6.93	*p* = 0.0002
APQ3 [%]	1.82 ± 1.64	5.81 ± 6.95	5.57 ± 7.15	*p* = 0.0001
APQ5 [%]	2.14 ± 2	6.29 ± 7.48	6.09 ± 8.02	*p* = 0.0006

## Data Availability

All the most relevant data generated or analyzed during this study are included in this published article and its Appendix A. Remaining datasets used and/or analyzed during the current study are available from the corresponding author upon reasonable request.

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
