# Peer review of "High-Speed Videoendoscopy Enhances the Objective Assessment of Glottic Organic Lesions: A Case-Control Study with Multivariable Data-Mining Model Development"

_cancers, 2023, doi:10.3390/cancers15143716_

Round 1
Reviewer 1 Report
Line 66 - I disagree with the term blurred image. With asynchronous vibration the image on LVS is not blurred but demonstrates irregular vibration
It would be good to know the types of benign vocal fold images that were assessed. Polyps, reinke's oedema, scar, nodules and cysts will all have different vibratory characteristics based upon which plane of the trilaminar vocal fold structure they are affecting. Granulomas are not relevant in thei study as they to typically not affect the mid portion of the vocal fold bvibration which was examined in this series
It would be good to know the type of vocal fold malignancies were assessed. Was this invasive SCC vs dysplasia. Also the vibratory characteristics of early glottic cancer once again depend entirely on how deep the lesion is invading. Early invasive cancer is less likely to cause severe mucosal wave interruption compared with disease extending down to the vocal ligament. I'm not sure that high speed would offer much benefit in more superficial cases.
Please define G1 and G2 SCC
The limitations of high speed imaging should be mentioned in this study. i.e. limited commercial availability in clinical practice, expense, time taken to perform analysis. Also how many participants were excluded because of inability to obtain suitable recordings?
Author Response
Reviewer 1
Thank you for your remarks concerning our research.
Line 66 - I disagree with the term blurred image. With asynchronous vibration the image on LVS is not blurred but demonstrates irregular vibration
You are entirely right, the term blurred image we used is not a precise one. We made the necessary corrections in the manuscript:
The resultant image of irregular vocal fold vibrations, which makes the assessment of certain vibratory features impossible, like in static light endoscopy. (Lines 66-67)
It would be good to know the types of benign vocal fold images that were assessed. Polyps, reinke's oedema, scar, nodules and cysts will all have different vibratory characteristics based upon which plane of the trilaminar vocal fold structure they are affecting. Granulomas are not relevant in thei study as they to typically not affect the mid portion of the vocal fold bvibration which was examined in this series
During conceptualization of the study, we made a decision in which types of vocal lesions machine learning would be of clinical use. We were aware that most kinds of lesions are very easy to differentiate based on white light endoscopy alone. That’s why we decided to include polyps, nodules and cysts. In our clinical experience those lesions tend to be confused by less experienced physicians. Also polyps and cystic lesions can hide early malignancy, detected only by careful kymographic examination by experienced ENT physician. Our concept was to use objective assessment to detect signs of malignancy, as it eliminates shortcomings of qualitative, subjective assessment, being able to help clinicians without broad experience and help the experienced otolaryngologists in doubtful cases. The largest group in our study were polyps (50 out of 64 subjects) and 14 patients had vocal cysts. We corrected the manuscript in order to precisely state what types of lesions were included in the benign group:
Among subjects with dysphonia 64 were diagnosed with benign vocal lesions (specifically: 50 polyps and 14 vocal cysts). (Lines 103-105)
It would be good to know the type of vocal fold malignancies were assessed. Was this invasive SCC vs dysplasia. Also the vibratory characteristics of early glottic cancer once again depend entirely on how deep the lesion is invading. Early invasive cancer is less likely to cause severe mucosal wave interruption compared with disease extending down to the vocal ligament. I'm not sure that high speed would offer much benefit in more superficial cases.
Histopathologic examination of patients with malignant lesions revealed invasive squamous cell carcinoma T1, in some cases accompanied with high grade dysplasia. There were no patients with only high grade dysplasia (carcinoma in situ) in this group. We completely agree that vibratory disturbance depend on depth of invasion. That is partially why we decided to use temporal parameters (describing irregularities in amplitude and frequency of oscillations in function of time) instead of spatial parameters (OpenQuotient, Amplitude and Asymmetry parameters). We found that in mild disruptions of mucosal wave, analysis of spatial disruptions in one cycle can be insufficient. On the other hand, quantitative assessment of hundreds of cycles reveal disruptions caused even by small lesions of the vocal folds. The next question we tried to answer in our study was: if there exist significant differences between disruptions caused by presence of benign and malignant lesion. Corrected part of the manuscript included below the next section.
Please define G1 and G2 SCC
Thank you for this remark, we made corrections to the manuscript: All of the organic lesions were confirmed by histopathologic examination. All malignancies were Grade 1 (well differentiated) or Grade 2 (moderately differentiated) invasive squamous cell carcinoma T1, in some cases accompanied by high grade dysplasia. Clinically the lesions were either exophytic tumors or infiltrative growths encompassing one vocal fold classified as cT1. (Line 108-113)
The limitations of high speed imaging should be mentioned in this study. i.e. limited commercial availability in clinical practice, expense, time taken to perform analysis. Also how many participants were excluded because of inability to obtain suitable recordings?
Thank you for this remark. We described the limitations in another of our work, mentioned in line 133:
Malinowski, J.; Niebudek-Bogusz, E.; Just, M.; Morawska, J.; Racino, A.; Hoffman, J.; BaraÅ„ska, M.; Kowalczyk, M.M.; Pietruszewska, W. Laryngeal High-Speed Videoendoscopy with Laser Illumination: A Preliminary Report. Otolaryngologia Polska 2021, 75, 1–10, doi:10.5604/01.3001.0015.2575
In this publication we describe precisely the setup we use and the improvements it makes compared to previously available HSV devices. Performing examination takes no more time than regular rigid optics endoscopy, during insertion of the endoscope there is real time preview available and captured high speed videos take from 10 – 100s to view fully. As to the analysis – performing full set of HSV derived analysis is semi-automatic and for an experienced operator takes about 10 minutes. As to the commercial availability: the total cost of our setup is comparable to a high class rigid endoscope unit with camera and stroboscope. To sum up we applied a HSV unit that has (to a certain degree) surpassed previous limitations.
As to the exclusions: there were only 5 such exclusions – all of them in malignant lesion group, mostly related to insufficient visibility of glottal gap during phonation, thus it was impossible to generate glottal with waveform.
Reviewer 2 Report
The authors are to be praised for their efforts in conducting this research. Implementing a machine learning approach to analyse data extracted from high-speed videoendoscopy is indeed an innovative method in laryngology. Your methods are robust and the choice to perform multivariable analysis is also correct as to avoid potential confouders among the three groups. Unfortunately, the fundamental flaw of your work is that lacks any clinical utility: by definition, euphonic versus dysphonic patients are differentiated only by clinical examination plus PROMs. HSV (as other additional endoscopic techniques such as stoboscopy or NBI etc.) have a role in helping the surgeon to customize the treatment in case small equivocal lesions (whatever their biological nature) are identified. In your paper, a clear description of exactly what lesions have been included in the benign and malignant group is lacking and ENT do not really need HSV to discriminate a cT2 carcinoma from a vocal fold cyst. This is a major limit of our methods which I would see useful only in the setting of leukoplakias and alike lesions.
Some minor suggestions for you: the bioinformatic methods may be shortened and left into an appendix; once an acronym is explicited, keep using it; some bibliographic entries (10 and 11) are badly reported
Some senteces are just wrong... An example: "The control group were 38 normophonic subjects without vocal fold pathology hos- 109 pitalized for other otolaryngological diseases.", to be corrected as ". The control group was composed of 38 normophonic subjects without vocal fold pathology on clincal examination and who had been hospitalized for other otolaryngological conditions." ...
Author Response
The authors are to be praised for their efforts in conducting this research. Implementing a machine learning approach to analyse data extracted from high-speed videoendoscopy is indeed an innovative method in laryngology. Your methods are robust and the choice to perform multivariable analysis is also correct as to avoid potential confouders among the three groups.
Unfortunately, the fundamental flaw of your work is that lacks any clinical utility: by definition, euphonic versus dysphonic patients are differentiated only by clinical examination plus PROMs.
Indeed, we employ widely used questionnaires in patients with voice disorders in our clinic. Functional outcomes are of utmost importance for us, and surveys help us in treatment monitoring. However we believe this theme is too extensive to include in this publication as we wanted to present a different approach.
We corrected fragment in our manuscript in order to emphasise the use of PROMs monitoring.
First all patients were assessed by ENT specialist. The examination included a complex interview concerning laryngeal symptoms. Each subject was required to complete voice quality questionnaires, as monitoring patients’ functional outcomes is of utmost importance in laryngological practice, it can also facilitate quicker detection of voice disorders.[25] To this purpose we employ validated surveys, among others Voice Handicap Index (VHI), Voice-Related Quality Of Life (V-RQOL), Vocal Tract Discomfort Scale (VTDS), and Voice Fatigue Index (VFI). (lines 124-131)
HSV (as other additional endoscopic techniques such as stoboscopy or NBI etc.) have a role in helping the surgeon to customize the treatment in case small equivocal lesions (whatever their biological nature) are identified. In your paper, a clear description of exactly what lesions have been included in the benign and malignant group is lacking and ENT do not really need HSV to discriminate a cT2 carcinoma from a vocal fold cyst. This is a major limit of our methods which I would see useful only in the setting of leukoplakias and alike lesions.
We agree that HSV has an important role in planning the surgery, we always employ both HSV and NBI in pre-operative assessment. Also it is true that for experienced ENT specialist it is possible to discriminate different types of vocal lesions with high confidence. At the same time we strongly believe that employment of advanced visual techniques has a very important clinical role, especially in case of less experienced physicians who may have difficulties in differentiating e.g. large, fibrotic polyps from early carcinomas. (quantitative approach in this case is crucial, as subjective kymographic interpretation causes many difficulties). It can also be of assistance to the experienced clinicians in doubtful cases by supplying objective parameters. Vibratory disturbances depend on depth of invasion. In our previous study concerning laryngotopographic evaluation of HSV, we proposed the Stiffness Asymmetry Index comparing quantitatively healthy and organically affected vocal fold. Our results showed that these glottal pathologies might be noninvasively, objectively distinguished prior to histopathological examination, showing clinical utility of HSV-derived visual techniques:
Kaluza, J.; Niebudek-Bogusz, E.; Malinowski, J.; Strumillo, P.; Pietruszewska, W. Assessment of Vocal Fold Stiffness by Means of High-Speed Videolaryngoscopy with Laryngotopography in Prediction of Early Glottic Malignancy: Preliminary Report. Cancers (Basel) 2022, 14, 4697, doi:10.3390/cancers14194697.
In this study we decided to use temporal parameters instead, as quantitative assessment of hundreds of cycles reveal disruptions caused even by small lesions of the vocal folds. The next question we tried to answer in our study was: if there exist significant differences between disruptions caused by presence of benign and malignant lesions. Machine learning answered that question positively and with high accuracy.
However, the description of lesions in our study was lacking. Among benign lesions the largest group were polyps (50 out of 64 subjects) and 14 patients had vocal cysts. We also improved the description of malignant cases.
Corrected fragments: Among subjects with dysphonia 64 were diagnosed with benign vocal lesions (specifically: 50 polyps and 14 vocal cysts). This group consisted of 44 women and 24 men, aged from 22 to 74 years, the average patient age was 53.62 ± 12.79 years. The remaining 36 subjects were diagnosed with early glottic cancer – out of them 27 were male and 9 were female, aged from 42 to 76 years, with an average age of 67.22 ± 6.66 years. All of the organic lesions were confirmed by histopathologic examination. All malignancies were Grade 1 (well differentiated) or Grade 2 (moderately differentiated) invasive squamous cell carcinoma, in some cases accompanied by high grade dysplasia. Clinically the lesions were either exophytic tumors or infiltrative growths encompassing one vocal fold classified as cT1. (Lines 103-113)
Some minor suggestions for you: the bioinformatic methods may be shortened and left into an appendix; We firmly believe that the bioinformatic method section is of utmost importance for comprehending the manuscript. Therefore, despite the suggestion made, we have decided to retain it within the main text. Our intention is to ensure that the section remains as concise as possible while retaining its significance.
once an acronym is explicited, keep using it;
We checked the consistence of acronym used and made the necessary corrections
some bibliographic entries (10 and 11) are badly reported – We corrected the bibliographic entries.
Comments on the Quality of English Language
Some senteces are just wrong... An example: "The control group were 38 normophonic subjects without vocal fold pathology hos- 109 pitalized for other otolaryngological diseases.", to be corrected as ". The control group was composed of 38 normophonic subjects without vocal fold pathology on clincal examination and who had been hospitalized for other otolaryngological conditions." ...
Thank you for this remark. We made the necessary corrections in the manuscript.
Reviewer 3 Report
The authors described the high speed stroboscopy of vocal cord end analyses of them. The authors ignored the kymography principes described in literature before like Svec et al. . The methodology and results described in paper are with more differences.
The results must be correlate to the literature more better and compare to the literature data exactly.
i am affraid that the rewriting must be done.
Author Response
The authors described the high speed stroboscopy of vocal cord end analyses of them. The authors ignored the kymography principes described in literature before like Svec et al. . The methodology and results described in paper are with more differences.
Thank you for your critical opinion of our research. Your remarks significantly helped us to improve our paper. As mentioned in our previous publication, during image acquisition we firstly visualize and assess the larynx, with the camera operating at around 24 fps. It allows the initial identification and centering the camera sensor on the glottis. Subsequently, the High-Speed mode is used, capturing images from the central part of the sensor. For the purpose of the study, images were recorded at a rate of 3200 fps. The length of the High-Speed recording was 2000 frames resulting in recording time of 625 ms. Next, we always perform kymographic analysis according to golden standards of kymography principles according to Svec. After recording phonatory vocal function with HSV camera, we generate kymograms in the middle part of the glottis. Later those kymograms are analyzed by ENT specialists in order to assess qualitative features as described by Svec at al. (1), including mucosal wave, amplitude and asymmetry. We included fragments of this analysis in Figure 1 – Subjects description. This is performed for every patient in our clinic, however amount of data gathered for each patient exceeds the scope of this paper. Additionally as noticed by Powell et al. (2) interrater agreement in case of those features analysis tends to differ basing on experience of the clinicians performing the rating. In this study we attempted to find an objective tool able to assist those of lesser experience in the field – facilitating detection and comparison of disruptions caused by both benign and malignant lesions, without relying heavily on examiner’s proficiency. In this study we decided to use temporal parameters, as quantitative assessment of hundreds of cycles reveal disruptions caused even by small lesions of the vocal folds. Two main groups of parameters were calculated: the Jitter group which describes changes in the frequency of oscillations in function of time and includes period perturbation measures and the Shimmer group, parameters which are related to changes in amplitude of vocal cycle during phonation and are called amplitude perturbation measures.
- Švec, J. G., & Schutte, H. K. (2012). Kymographic imaging of laryngeal vibrations. Current Opinion in Otolaryngology & Head and Neck Surgery, 20(6), 458–465. doi:10.1097/moo.0b013e3283581feb
- Powell, M. E., Deliyski, D. D., Hillman, R. E., Zeitels, S. M., Burns, J. A., & Mehta, D. D. (2016). Comparison of Videostroboscopy to Stroboscopy Derived From High-Speed Videoendoscopy for Evaluating Patients With Vocal Fold Mass Lesions. American Journal of Speech-Language Pathology, 25(4), 576. doi:10.1044/2016_ajslp-15-0050
The results must be correlate to the literature more better and compare to the literature data exactly.
i am affraid that the rewriting must be done.
Unfortunately, as mentioned in the discussion, qualitative HSV analysis in glottic organic lesions is a relatively new topic, and amount of available literature is limited, as mentioned in the manuscript (Lines 293-299). It is also known that while HSV has a potential for clinical adaptation, it lacks standardized investigation protocol and analysis tools. Attempts have been made to create such instruments, however the number of available cases was limited:
Kist, A. M., Dürr, S., Schützenberger, A., & Döllinger, M. (2021). OpenHSV: an open platform for laryngeal high-speed videoendoscopy. Scientific Reports, 11(1). doi:10.1038/s41598-021-93149-0
We made an attempt at pre-operative differentiation of organic lesions based on the temporal parameters describing vocal fold oscillations. We referred our results in comparison to studies by other authors in available source literature.
Yamauchi, A.; Imagawa, H.; Yokonishi, H.; Sakakibara, K.-I.; Tayama, N. Multivariate Analysis of Vocal Fold Vibrations on Var-ious Voice Disorders Using High-Speed Digital Imaging. Applied Sciences 2021, 11, 6284, doi:10.3390/app11146284
Noordzij, J.P.; Woo, P. Glottal Area Waveform Analysis of Benign Vocal Fold Lesions before and after Surgery. Annals of Otology, Rhinology and Laryngology 2000, 109, 441–446, doi:10.1177/000348940010900501.
Gandhi, S.; Bhatta, S.; Ganesuni, D.; Ghanpur, A.D.; Saindani, S.J. High-Speed Videolaryngoscopy in Early Glottic Carcinoma Patients Following Transoral CO2 LASER Cordectomy. European Archives of Oto-Rhino-Laryngology 2021, 278, doi:10.1007/s00405-020-06433-6.
Reviewer 4 Report
Good work. A little mention about the use of questionnaire in the study of these patients must be done.
Please, you should mention in the text and in the references this work :
Galletti B, Sireci F, Mollica R, Iacona E, Freni F, Martines F, Scherdel EP, Bruno R, Longo P, Galletti F. Vocal Tract Discomfort Scale (VTDS) and Voice Symptom Scale (VoiSS) in the Early Identification of Italian Teachers with Voice Disorders. Int Arch Otorhinolaryngol. 2020 Jul;24(3):e323-e329. doi: 10.1055/s-0039-1700586. Epub 2019 Dec 13. PMID: 32754244; PMCID: PMC7394657.
Author Response
Good work. A little mention about the use of questionnaire in the study of these patients must be done.
Please, you should mention in the text and in the references this work :
Galletti B, Sireci F, Mollica R, Iacona E, Freni F, Martines F, Scherdel EP, Bruno R, Longo P, Galletti F. Vocal Tract Discomfort Scale (VTDS) and Voice Symptom Scale (VoiSS) in the Early Identification of Italian Teachers with Voice Disorders. Int Arch Otorhinolaryngol. 2020 Jul;24(3):e323-e329. doi: 10.1055/s-0039-1700586. Epub 2019 Dec 13. PMID: 32754244; PMCID: PMC7394657.
Thank you for your appreciation of our work. Indeed we use questionnaires in all patients treated in our clinic. Functional outcomes are of utmost importance for us, and surveys help us in treatment monitoring. To this goal we employ validated surveys, among others Voice Handicap Index (VHI), Voice-Related Quality Of Life (V-RQOL), Vocal Tract Discomfort Scale (VTDS) and Voice Fatigue Index (VFI).
We corrected this fragment in our manuscript and cited the paper mentioned above.
First all patients were assessed by ENT specialist. The examination included a complex interview concerning laryngeal symptoms. Each subject was required to complete voice quality questionnaires, as monitoring patients’ functional outcomes is of utmost importance in laryngological practice, it can also facilitate quicker detection of voice disorders.[25] To this purpose we employ validated surveys, among others Voice Handicap Index (VHI), Voice-Related Quality Of Life (V-RQOL), Vocal Tract Discomfort Scale (VTDS), and Voice Fatigue Index (VFI).
Reviewer 5 Report
The authors performed prospective study of 100 patients with unilateral vocal fold lesions and 38 controls using high-speed videoendoscopy (HSV). They demonstrated that machine learning established the diagnostic tool with HSV parameters to distinguish vocal fold lesions from the normal. Interestingly, they presented the machine learning model to find malignancies among organic lesions with high accuracy. They showed possibility of the machine learning models of HSV for future practical use.
Only a minor issue should be checked.
line 103, 44 women + 24 men = 68. Not 64.
This is a very practical research, giving head and neck oncologists hints to use machine learning.
Author Response
Reviewer 5
Comments and Suggestions for Authors
The authors performed prospective study of 100 patients with unilateral vocal fold lesions and 38 controls using high-speed videoendoscopy (HSV). They demonstrated that machine learning established the diagnostic tool with HSV parameters to distinguish vocal fold lesions from the normal. Interestingly, they presented the machine learning model to find malignancies among organic lesions with high accuracy. They showed possibility of the machine learning models of HSV for future practical use.
Only a minor issue should be checked.
line 103, 44 women + 24 men = 68. Not 64.
Thank you for appreciation of our work and for finding this small counting mistake made during revisions. We corrected it in the manuscript.
Round 2
Reviewer 2 Report
I understand yet do not fully agree with your comments
Author Response
Dear Reviewer,
We would like to thank you very much for your effort and time devoted to reviewing our manuscript and for your kind words and positive feedback on our research. We have completely rearranged our conclusions, because we fully agree that the standard protocol for distinguishing between benign and malignant lesions continues to be clinical evaluation with videolaryngoscopy, executed by an experienced ENT specialist and finally confirmed by histopathological examination.
Corrected fragments:
In conclusion, our study provides a comprehensive analysis of the predictive capabilities of audio and HSV data in the assessment of vocal fold oscillations. The standard protocol for distinguishing between benign and malignant lesions continues to be clinical evaluation with videolaryngoscopy, executed by an experienced ENT specialist and finally confirmed by histopathological examination. Our study did not establish that any single parameter, derived from HSV kymographic analysis, could differentiate noncancerous from malignant laryngeal lesions. However, our findings did suggest that advanced machine learning models, which consider the complex interactions present in HSV data, could potentially indicate a heightened risk of malignancy. Therefore, this technology could prove pivotal in aiding early cancer detection, thereby emphasizing the need for further investigations and validations. (Lines 411-421)
Reviewer 3 Report
The conclusions are inadequate. The diagnose of benigne or malignant disease due by voice analyses include. All of desribed methods is not possible and the therapy of malignant disease based on the results of vocal cord function using AI is unable to accept.
for that must be the authors correct the results and apply the clear results for this conslusion or reject this paper.
the qustion is what will be do the AI and voice test for patients with any form of RNS or RNL compare the results of this paper.
the conclusions are bad and the results are not based on results and based on the inadeqate to desribed resulrs.
Author Response
Dear Reviewer,
Thank you very much for your thoughtful comments and efforts toward improving our manuscript.
We have completely rearranged our conclusions, because we fully agree that the standard protocol for distinguishing between benign and malignant lesions continues to be clinical evaluation with videolaryngoscopy, executed by an experienced ENT specialist and finally confirmed by histopathological examination.
Corrected fragments:
The standard protocol for distinguishing between benign and malignant lesions remains clinical judgment and histopathological confirmation by an experienced otolaryngologist. Additional tool is high-speed videoendoscopy (HSV) being accurate method for objective assessment of the vocal fold oscillations. The aim of the study was to utilize quantitative assessment of the vibratory characteristics of vocal folds in diagnosing benign and malignant lesions of the glottis using HSV. The machine learning model identifying malignancy among organic lesions reached AUC equal to 0.85 and presented 80.6% accuracy, 100% sensitivity, and 71.1% specificity on the training set and important predictive factors were frequency perturbation measures. The results did suggest that advanced machine learning models basing on HSV analysis, could potentially indicate a heightened risk of cancerous mass. Therefore, this technology could in the future aid early cancer detection, however further investigations and validations are needed. (Lines 14-24)
The standard protocol for distinguishing between benign and malignant lesions continues to be clinical evaluation by an experienced ENT specialist and confirmed by histo-pathological examination. Our findings did suggest that advanced machine learning models, which consider the complex interactions present in HSV data, could potentially indicate a heightened risk of malignancy. Therefore, this technology could prove pivotal in aiding early cancer detection, thereby emphasizing the need for further investigations and validations. (Lines 36-42)
Conclusions
In conclusion, our study provides a comprehensive analysis of the predictive capabilities of audio and HSV data in the assessment of vocal fold oscillations. The standard protocol for distinguishing between benign and malignant lesions continues to be clinical evaluation with videolaryngoscopy, executed by an experienced ENT specialist and finally confirmed by histopathological examination. Our study did not establish that any single parameter, derived from HSV kymographic analysis, could differentiate noncancerous from malignant laryngeal lesions. However, our findings did suggest that advanced machine learning models, which consider the complex interactions present in HSV data, could potentially indicate a heightened risk of malignancy. Therefore, this technology could prove pivotal in aiding early cancer detection, thereby emphasizing the need for further investigations and validations. (Lines 411-421)
Round 3
Reviewer 3 Report
The authors improved text a little bit. The main idea that the voice analyses can be significant to diagnose of malignant or benigne tumors is optimistic but not relevant ti clinical practice yet. The authors oversize the idea to results and method described in his paper generally. I would like recommand to the authors the revision of that for the bettter scince soudness of the work with good and wxcat voice and VHS analyse. The authors results are very interesting for the clinical practice if the intepretation will be associate to the better reconize of vocal cord infiltration by pathological tissue but the differenece between the malignant and bnigne cells is very oprimistic and nor relevant. If the authors will focused on the deep of infiktration Surface or through the all mucouse or throu the laryngeal muscules) the resuktc will be more usefull in clinical practice. The main idea for any is the telemdicine and the diagnose by phones. This idea is not based on relevant data for the decision of therapy yet. The more better study designe is noted to that in future.